# Association between Galectin Levels and Neurodegenerative Diseases: Systematic Review and Meta-Analysis

**DOI:** 10.3390/biom12081062

**Published:** 2022-07-31

**Authors:** Edgar Ramos-Martínez, Iván Ramos-Martínez, Iván Sánchez-Betancourt, Juan Carlos Ramos-Martínez, Sheila Irais Peña-Corona, Jorge Valencia, Renata Saucedo, Ericka Karol Pamela Almeida-Aguirre, Marco Cerbón

**Affiliations:** 1Facultad de Química, Universidad Nacional Autónoma de México, Coyoacán 04510, Mexico; edgarramos@ciencias.unam.mx (E.R.-M.); sheila.ipc@live.com (S.I.P.-C.); karolalmeida@ciencias.unam.mx (E.K.P.A.-A.); 2Departamento de Medicina y Zootecnia de Cerdos, Facultad de Medicina Veterinaria y Zootecnia, Universidad Nacional Autónoma de México, Coyoacán 04510, Mexico; iramos.martinez88@gmail.com (I.R.-M.); aisb_7@yahoo.com.mx (I.S.-B.); 3Departamento de Cardiología, Hospital General Regional Lic Ignacio Garcia Tellez IMSS, Cuauhtémoc 97150, Mexico; dr.juancarlosramos@hotmail.com; 4Unidad de Investigación en Endocrinología, UMAE Hospital de Especialidades, Instituto Mexicano del Seguro Social, Cuauhtémoc 06720, Mexico; j.valencia.o@hotmail.com (J.V.); sgrenata@yahoo.com (R.S.); 5Unidad de Investigación en Reproducción Humana, Instituto Nacional de Perinatología “Isidro Espinosa de los Reyes”—Facultad de Química, Universidad Nacional Autónoma de México, Mexico City 04510, Mexico

**Keywords:** Alzheimer’s disease, amyotrophic lateral sclerosis, galectin, multiple sclerosis, neurodegenerative diseases, Parkinson’s disease

## Abstract

Galectins are a family of proteins with an affinity for β-galactosides that have roles in neuroprotection and neuroinflammation. Several studies indicate that patients with neurodegenerative diseases have alterations in the concentration of galectins in their blood and brain. However, the results of the studies are contradictory; hence, a meta-analysis is performed to clarify whether patients with neurodegenerative diseases have elevated galectin levels compared to healthy individuals. Related publications are obtained from the databases: PubMed, Central-Conchrane, Web of Science database, OVID-EMBASE, Scope, and EBSCO host until February 2022. A pooled standard mean difference (SMD) with a 95% confidence interval (CI) is calculated by fixed-effect or random-effect model analysis. In total, 17 articles are included in the meta-analysis with a total of 905 patients. Patients with neurodegenerative diseases present a higher level of galectin expression compared to healthy individuals (MDS = 0.70, 95% CI 0.28–1.13, *p* = 0.001). In the subgroup analysis by galectin type, a higher galectin-3 expression is observed in patients with neurodegenerative diseases. Patients with Alzheimer’s disease (AD), amyotrophic lateral sclerosis (ALD), and Parkinson’s disease (PD) expressed higher levels of galectin-3. Patients with multiple sclerosis (MS) have higher levels of galectin-9. In conclusion, our meta-analysis shows that patients with neurovegetative diseases have higher galectin levels compared to healthy individuals. Galectin levels are associated with the type of disease, sample, detection technique, and region of origin of the patients.

## 1. Introduction

Neurodegenerative diseases affect millions of people worldwide. These diseases are characterized by the gradual development of cognitive dysfunction and motor impairment caused by neuronal damage and death. The most common neurodegenerative diseases are Alzheimer’s disease (AD), Parkinson’s disease (PD), amyotrophic lateral sclerosis (ALS), multiple sclerosis (MS), and Huntington’s disease (HD) [1]. Neurodegenerative diseases are not curable due in part to a lack of understanding of pathogenesis and treatments starting at late stages [2], so the search for diagnostic and prognostic markers is important for treatment.

Among the proteins that have been investigated in neurodegenerative diseases, we have galectins, which are a group of lectins with an affinity for structures with β-galactoside [3]. Galectins can be localized in different compartments such as the nucleus, cytoplasm, and extracellular space, and their functions depend on the compartment where they are localized [4,5,6]. Seventeen types of galectins have been described, of which the most studied are galectin-1 and galectin-3. Galectins are involved in processes such as cell cycle, RNA transcription, cell differentiation, and immune response [3,7].

Alterations in galectin expression have been observed in patients with neurodegenerative diseases [8,9,10]; furthermore, galectins mediate functions related to neurodegeneration and neuroinflammation [11,12,13,14]. For example, galectins mediate cell-cell or cell-matrix interactions, cell growth, differentiation, apoptosis, and activation of immune system cells such as macrophages and microglia [12]. Galectins form clusters known as lattices that stabilize extracellular matrix proteins, organize receptors on the cell membrane and partly control intracellular trafficking [3]. In the case of galectin-4, it participates in lipid raft-dependent secretion through glycoprotein-lectin interactions, and galectin-3 mediates the secretion of proteins not associated with lipid rafts [3].

Astrocytes express galectin-1, which is involved in the proliferation of adult neural stem cells [5]. Galectin-1 also mediates the activation of regulatory T cells and thus participates in the protection of axonal damage [5]. Some populations of neurons also express galectins, which have been implicated in pain control [5]. In the case of galectin-3, it has been observed that its expression is associated with microglia activation in brain damage, ischemia, demyelination, and encephalitis [5,12]. Activated microglia release galectin-3, which activates other microglial cells, facilitates phagocytosis of neurons, and promotes beta-amyloid aggregation [13,14]. In addition, galectin-3 increases interleukin (IL)-1β expression in microglia from a murine model of Huntington’s disease [14]. In primary cultures of microglia, suppression of galectin-3 expression enhances the elimination of damaged lysosomes [14], and when galectin = 3 expression is eliminated in mice with experimental autoimmune encephalomyelitis (EAE), the severity of the disease is reduced [12].

Given the involvement of galectins in neurodegeneration-related processes, galectin levels have been measured in plasma, cerebrospinal fluid, and the brains of patients with neurodegenerative diseases. However, the results of studies evaluating the level of galectins in neurodegenerative diseases are contradictory; therefore, the aim of this meta-analysis is to evaluate galectin expression levels in comparison with healthy individuals, in addition to determining possible sources of heterogeneity.

## 2. Materials and Methods

### 2.1. Identification of Relevant Literature

This systematic review was reported following the Preferred Reporting Items for Systematic Reviews and Meta-Analyses (PRISMA) statement [15]. The protocol of the present systematic review was registered on PROSPERO (CRD42022316539). The search was performed in the following databases: PubMed, Central-Cochrane, Wed of Science database, OVID-EMBASE, Scope, and EBSCOhost, and in search engines such as Google Scholar. The following combined terms were searched: (Galectin or Gal or LGALS or S-type lectin or galactose binding lectin) and (neurodegenerative diseases or Parkinson’s disease or Alzheimer’s disease or Huntington’s disease or multiple sclerosis or epilepsy or amyotrophic lateral sclerosis). The last search was performed in February 2022.

Two coauthors (Ramos-Martinez, Edgar, and Ramos-Martinez, Ivan) independently reviewed all abstracts generated by the search and read the full text to identify eligible studies and potentially eligible articles were identified by searching reference lists of relevant reviews and original articles. Studies in English and Spanish with full access to the article were included, and no restriction on publication period was applied.

### 2.2. Inclusion and Exclusion Criteria

The inclusion criteria in this meta-analysis were (a) patients were diagnosed with neurodegenerative diseases such as Parkinson’s disease, Alzheimer’s disease, Huntington’s disease, multiple sclerosis, epilepsy or amyotrophic lateral sclerosis, (b) galectin expression was determined by immunohistochemistry, ELISA, Western blot, PCR, mass spectrometry, (c) the level of galectin expression was compared with healthy individuals, and (d) galectin levels were measured in serum, plasma, brain or other tissue. Studies were excluded according to the following criteria: (a) duplicate studies from the original data, (b) abstracts, comments, case reports, unpublished data, letters, reviews, or nonclinical studies, and (c) the research without access to full text.

### 2.3. Quality Evaluation for Studies

Study quality was assessed using the Newcastle–Ottawa quality assessment scale. We evaluated the studies according to the following three fundamental criteria established by the scale: (1) the selection of the study groups, (2) the comparability of the groups, and (3) the ascertainment of either the outcome or the exposure. Studies with scores greater than 7 are considered high quality, a study could be awarded a maximum of nine points.

### 2.4. Data Extraction

We used a data extraction sheet to extract data from the included studies to assess study quality and evidence synthesis. Data from selected articles were extracted independently by two reviewers (Ramos-Martínez, E.G., and Ramos-Martínez, I.E.), and discrepancies were identified and resolved by discussion with a third reviewer (Cerbón, M.).

### 2.5. Statistical Analysis

Statistical analysis in this meta-analysis was performed using Review Manager (RevMan) and Comprehensive Meta-Analysis software. The mean and standard deviation of the relative level of galectin expression in arbitrary units were obtained. In studies reporting galectin levels through median and range, the formulas specified by Hozo et al. (2005) [16] were used to obtain the mean and standard deviation. In the articles that did not report the mean and standard deviation, the mean and standard deviation were obtained from the graphs with the WebPlotDigitizer program [17]. Finally, the standardized mean difference (SMD) was calculated from these data.

We used the I^2^ test and the χ^2^-based Q test to analyze the statistical heterogeneity between studies. When there was significant heterogeneity (I^2^ > 50% or *p* < 0.05) in the included studies, the random-effects model was then calculated according to the DerSimonian–Laird method [18]. Otherwise, the fixed-effects model [19] was used. Subgroup analyses were performed to analyze sources of heterogeneity. Sensitivity analysis was also performed, in which one study at a time was eliminated to assess its impact on the results of the meta-analysis. Finally, bias analysis was performed using the funnel plot and Egger’s and Begg’s tests. The trim-and-fill method [20] was used to estimate the effect of publication bias in the calculation of the MDS and its 95% confidence interval.

## 3. Results

### 3.1. Characteristics of the Selected Studies and Qualitative Analysis

We found 2609 studies through the search strategy described above. After eliminating duplicate articles, we reviewed the titles and abstracts of 832 articles, of which 789 were excluded because they were not related to the object of study. The reviewers evaluated 43 full articles, and 21 were excluded due to the following reasons: 10 did not measure galectins, 2 were not performed with neurodegenerative diseases, 7 were reviews or conference abstracts, and 2 were performed in mice. Details of the search are presented in Figure 1.

Twenty-two articles were selected, of which 5 were addressed in the qualitative analysis and 17 were used in the quantitative analysis. In the articles for the qualitative analysis, three were in ALS patients, one in SM, and one in prion diseases. Studies in ALS indicate that galectin-1 is expressed in neurofilamentous lesions [21], and no differences in plasma levels of galectin-1 and galectin-3 were found between patients with rapid progression and patients with slow progression [10]. However, a reduction in galectin-1 expression in the skin was observed in ALS patients compared to patients with other neurodegenerative and inflammatory diseases. The study in SM found that galectin-4 is expressed in chronic lesions in the brain where it possibly participates in remyelination and differentiation functions of oligodendrocytes [22]. In the study of prion diseases, it was determined that galectin-1 levels in the brain are increased in patients, although the sample size was very small [23].

In the quantitative analysis, 905 patients with neurodegenerative diseases were included. In 12 of these studies, gal-3 is evaluated in 5 gal-1, in 3 gal-9, in one gal-4, and in one gal-8. Regarding the type of neurodegenerative disease, six studies were in Alzheimer’s disease, five in Parkinson’s disease, three in Amyotrophic lateral sclerosis (ALS), two in MS, one in HD, and one in intractable epilepsy. These studies were conducted in nine different countries; six were from Europe, six from East Asia, three from Western Asia, and one from America. In the chosen studies, galectin levels had different cohort levels and were mainly determined by ELISA, Western blot, RT-qPCR, and 2-DE and MALDI-TOF-MS (Table 1). We assessed the quality of the studies with the Newcastle-Ottawa Scale. We considered the study high quality if it had more than seven points on the scale. Only the studies by Tao et al. (2020) and Zhou et al. (2010) had a low quality. The complete evaluation data appears in Appendix A.

### 3.2. Galectin Expression Levels in Neurodegenerative Diseases

Of the 17 studies included in the meta-analysis, 4 studies reported the mean and standard deviation according to disease subtype [9,30,32,35], so we pooled the data from each article using the formulas recommended by the Cochrane Handbook for Systematic Reviews of Interventions [38] to obtain a pooled mean and standard deviation, and thus avoid the error of repeating a single control group for two or more patient subgroups of a study. Our results show that patients with neurodegenerative diseases present a higher level of galectin expression compared to healthy individuals (MDS = 0.70, 95% CI 0.28–1.13, *p* = 0.001; Figure 2). Study heterogeneity was high (I^2^ = 93%, *p* < 0.00001), so a random-effects model was used for analysis.

### 3.3. Heterogeneity Analysis

To explore the source of heterogeneity, a subgroup analysis was performed by type of neurodegenerative disease, type of galectin, region to which patients belong, method of galectin detection, and type of sample analyzed. Analysis according to neurodegenerative disease type showed that there was a significant difference in galectin expression in ALS (MDS = 0.31, 95% CI 0.04–0.59, *p* = 0.03) and MDS (MDS = 0.86, 95% CI 0.54–1.18, *p* < 0.00001), but not in AD and PD (Figure 3a). In the case of epilepsy and HD, there was only one study. Analysis by galectin type was performed for galectin-3, galectin-1, and galectin-9, which were those for which more than two studies were available. A significant difference in galectin-3 expression was found between patients with neurodegenerative diseases and healthy individuals (MDS = 0.85, 95% CI 0.53–1.18, *p* < 0.00001), but no differences were found for galectin-1 and 9 (Figure 3b).

Since different types of galectin were evaluated in the studies of patients with AD, PD, and SM, we performed a subgroup analysis considering only one specific type of galectin. In AD we considered the expression of Gal-3, in SM that of Gal-9, and in PD the expression of Gal-1 or Gal-3, since there were more than two studies for these galectins. We found that patients with AD and PD expressed more galectin-3 (MDS = 0.64, 95% CI 0.45–0.83 and MDS = 0.58, 95% CI 0.28–0.88, respectively) and patients with MS expressed more galectin-9 (MDS = 1.03, 95% CI 0.62–1.44) compared to healthy individuals (Table 2).

In the subgroup analysis by region, higher galectin expression was observed in patients with neurodegenerative diseases for the Western Asia and European group (MDS = 0.60, 95% CI 0.40–0.79 and MDS = 0.76, 95% CI 0.10–1.14, respectively). As for the detection method, higher galectin expression was found in patients with neurodegenerative diseases in the Western blot subgroup (SMD = 0.87, 95% CI 0.47–1.27, *p* < 0.0001). In the subgroup analysis by type of sample detected, it was found that studies analyzing brain samples report higher galectin expression in patients with neurodegenerative diseases (SMD = 0.93, 95% CI 0.57–1.30, *p* < 0.00001) (Table 2).

### 3.4. Sensitivity Analyses and Publication Bias

A sensitivity analysis was performed by removing one study at a time to assess the individual impact on the overall MDS estimate. Our results indicate that only the study by Tian et al. (2016) deviates somewhat from the data of the other studies, but since its weight in the meta-analysis is only 4.4% (Figure 2), it does not significantly influence SMD, nor does its 95% CI (Figure 4a).

Publication bias was explored using funnel plots and Egger’s and Begg’s tests. The funnel plot shows an asymmetric distribution of the data (Figure 4b), indicating a possible publication bias. This agrees with what was found when applying Egger’s test (t value = 2.59, *p* = 0.016). Begg’s test did not show significant publication bias (Tau = 0.183, *p* = 0.199). We applied the Trim and Fill method [20] to assess the effect of possible publication bias of studies with small. The adjusted values with the Trim and fill method do not vary significantly from those reported previously [SMD = 0.76, 95% CI 0.33–1.19]; moreover, the method does not suggest introducing additional studies for effect correction. This suggests that the asymmetry observed in the funnel plot may be due to the heterogeneity of the studies rather than to the effect of bias. The heterogeneity of the studies is related to the type of galectin, the type of neurodegenerative disease, and the type of sample analyzed, as shown in the subgroup analysis.

## 4. Discussion

Galectins have been linked to microglia activation, phagocytosis of neurons, formation of beta-amyloid or Tau protein aggregates, and induction of cytokines in the nervous system [35,39]. Several studies have shown alterations in galectin levels in models of neurodegenerative diseases [11,14,40,41]; furthermore, studies in patients indicate changes in galectin concentration in both the brain and peripheral blood [8,9,10]. Thus, our study aimed to clarify whether there are difference in galectin expression levels between patients with neurodegenerative diseases and healthy individuals and to detect sources of heterogeneity.

Our analysis shows that patients with neurodegenerative diseases express higher levels of galectin than healthy individuals. In the subgroup analysis by galectin type, we only found significant differences between patients and healthy individuals for galectin 3. In the subgroup analysis by type of neurodegenerative diseases and galectin, it was shown that patients with AD, PD, and ALS have higher levels of galectin-3 expression, and in the case of patients with MS, higher expression of gal 9 was found.

Only one human study was found that evaluated galectin levels in HD. This study showed an increase in galectin-3 in the brain and plasma of HD patients, in addition to which a positive correlation of galectin-3 expression with clinical manifestations of the disease and a negative correlation with the Mini-Mental State Exam (MMSE) score were observed [14]. In this meta-analysis, we included a study performed on intractable epilepsy, although the question of whether there is neurodegeneration in epilepsy is still open, there are currently data in temporal lobe epilepsy in humans and in murine models that support the existence of a neurodegenerative process in epilepsy [42,43]. The study by Tian et al. (2016) [31] found increased serum galectin-3 levels in intractable epilepsy compared to healthy individuals.

As can be seen, most studies point to an increase in the expression of galectins in neurodegenerative diseases, and most of these have focused on evaluating galectin-3 since it is the one for which the most function has been described in the nervous system. This increased expression of galectins in patients suggests their involvement in disease etiology. In the following, we discuss the main functions that have been described for galectins in neurodegenerative diseases.

Galectin-3 is expressed by astrocytes, microglia, macrophages, dendritic cells, and activated T and B cells. Inflammatory mediators such as IL-1 increase its expression [44]. In primary cultures of microglia and astrocytes, treatment with galectin-3 increases the expression of inflammatory cytokines such as TNF-α, IL-1β, IL-6, and IFN-γ, but not of anti-inflammatory cytokines such as TGF-β and IL-10 [45]. Galectin-3 promotes oligodendrocyte differentiation and contributes to remyelination by modulating microglia and matrix metalloprotease activity [46]. Galectin-3 is secreted by active microglia to stimulate other microglial cells through Toll-like receptor 4 (TLR4) and Triggering receptor expressed on myeloid cells 2 (TREM2) as part of a positive feedback mechanism [39]; furthermore, galectin-3 can opsonize neurons with low sialic acid content and mediate their phagocytosis [13]. In murine AD molds, galectin-3 is overexpressed and promotes beta-amyloid aggregation and its toxicity [11,35]. Suppression of galectin-3 in 5xFAD mice, a murine model of AD, improves cognitive performance, reduces expression in microglia of inflammatory genes, and reduces amyloid plaques [11].

In models of other neurodegenerative diseases, galectin-3 has also been linked to damage and neuroinflammation. For example, in R6/2 mice; a murine model of HD expressing exon-1 of the human huntingtin gene, galectin-3 is overexpressed in microglia and contributes to inflammation through NF-κB (Nuclear factor κB) and NLRP3 inflammasome-dependent pathways. Suppression of galectin-3 reduces inflammation and huntingtin aggregation, improves motor dysfunction, and increases survival in HD mice [14]. In the case of mice with experimental autoimmune encephalomyelitis (EAE), a model of multiple sclerosis, a reduction in disease severity is presented when galectin-3 expression is suppressed [47]. Therefore, it has been suggested that galectin-3 may be an indicator of prognosis, mortality, or remission in neurodegenerative diseases [12].

As for galectin-1, it has been shown to reduce the inflammatory response in LPS-activated neuroblastoma cells. In mice treated with MPTP (1-methyl-4-phenyl-1, 2, 3, 6-tetrahydropyridine), a PD model, it was reported that galectin-1 inhibited microglia activation, decreased secretion of interleukin (IL)-1, Tumor Necrosis Factor alpha (TNF-α), inducible nitric oxide synthase (iNOS) and cyclooxygenase-2 (COX-2) [48]. Furthermore, galectin-1 attenuates cognitive dysfunction and reduces surgery-induced hippocampal neuronal death in aged mice through decreasing microglial activation and expression of IL-1β, IL-6, and TNF-α. It has been proposed that galectin-1 reduces microglial activation by decreasing the translocation of NF-κB p65 and c-Jun [49]. Another mechanism through which galectin-1 suppresses the production of iNOS, TNF-α, and CCL2 in microglia is by binding to CD45, which favors their retention on the cell membrane [50]. Suppression of galectin-1 in an EAE model resulted in increased neurodegeneration, and the transfer of galectin-1-expressing astrocytes suppressed the development of EAE [50].

Another study showed that galectin-1 interacts with the Neuropilin-1/PlexinA4 receptor complex in damaged neurons to contribute to axonal regeneration and locomotor recovery after spinal cord injury [51]. In a rat model of ischemia, galectin-1 increases brain-derived neurotrophic factor (BDNF) expression in astrocytes. BDNF is a factor involved in neuronal survival and is essential for learning and memory [52]. Finally, galectin-1 promotes the proliferation of neurolanular progenitors in the hippocampus [41]. In the case of galectin-8, it has been reported to protect against the formation of tau aggregates by promoting autophagy in cells that have suffered damage to their endomembrane systems [53]. In cultured hippocampal neurons, galectin-8 protects neurons from glutamate-induced excitotoxicity, oxidative stress, and beta-amyloid oligomers [54]. In EAE, galectin-8 eliminates Th17 cells and ameliorates clinical manifestations [55].

Galectin-9 is expressed in several brain regions such as the cortex, olfactory bulb, hippocampus, thalamus, hypothalamus, amygdala, and cerebellum [56]. This galectin is not constitutively expressed in astrocytes, but its expression is induced by TNFα, IL-1β, or interferon γ (IFNγ) [57]. Galectin-9 expression is increased in microglia after stimulation with poly (I:C) [58]. Among the functions of galectin-9 related to neuroinflammation, we have found that it increases apoptosis in encephalitogenic T-cells [57] and increases the production of TNFα and IL-6 in microglia [58]. In inactive multiple sclerosis lesions, galectin-9 is localized in the cytoplasmic region, but in active lesions it is present in both the cytoplasm and the nucleus [9]. The functional relevance of the change in galectin-9 localization is not known, but there are reports that some galectins interact with nuclear factors [4].

In a model of intracerebral hemorrhage, a correlation was found between galectin-9 expression and the number of type 2 microglia, as well as an anti-inflammatory cytokine profile. Increased galectin-9 also correlates with improved motor, sensory and memory abilities, and decreased cell death [59]. Thus, there is ample evidence to support the functional role of galectins in neurodegenerative processes.

Some limitations of our study are the following: (1) publication bias, as there is a tendency to report positive results; (2) most studies have focused on galectin-3 and there are few studies on the other galectins; (3) different detection techniques for galectins were used in the included studies, which implies different sensitivity and specificity of the tests; (4) the type of sample evaluated; (5) the quality of the studies is variable and many had small samples.

Despite the limitations, our study shows alterations in galectin concentration in neurodegenerative diseases, which is supported by observations in murine models. Future studies need to evaluate other types of galectins and determine their cellular localization. It has already been suggested that the potential prognostic value of galectins depends on cellular localization [6,60,61].

## 5. Conclusions

Our study indicates that there is an increase in galectin expression in neurodegenerative diseases, pointing to involvement in neuronal damage and neuroinflammatory processes. An increase in galectin-3 was observed in AD, PD, and ALS. An increase in Gal-9 expression was observed in SM. Further studies are required, especially of galectins other than Gal-3, to determine whether there are alterations in their expression and implications in neurodegenerative diseases.

## Figures and Tables

**Figure 1 biomolecules-12-01062-f001:**
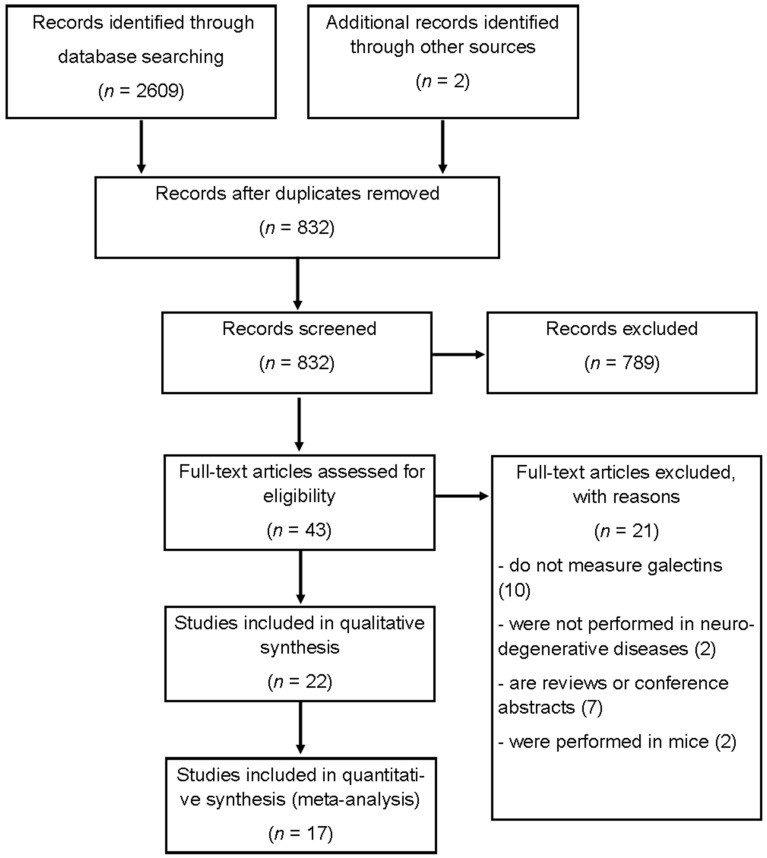
Flow diagram of the publication search and selection process.

**Figure 2 biomolecules-12-01062-f002:**
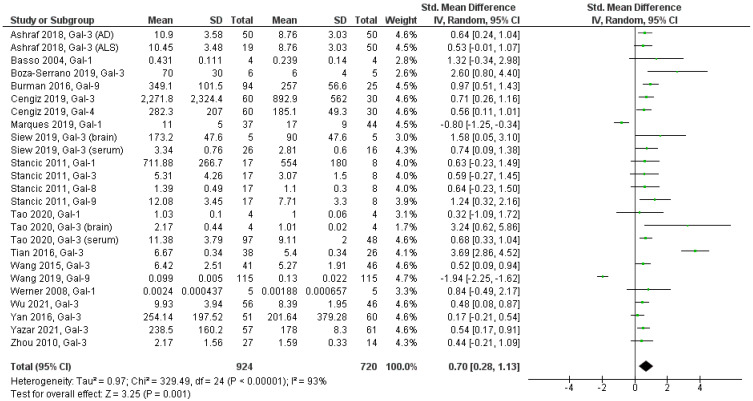
Forest plot showing the results of the meta-analysis of random effects of galectin levels in patients with neurodegenerative diseases and healthy individuals.

**Figure 3 biomolecules-12-01062-f003:**
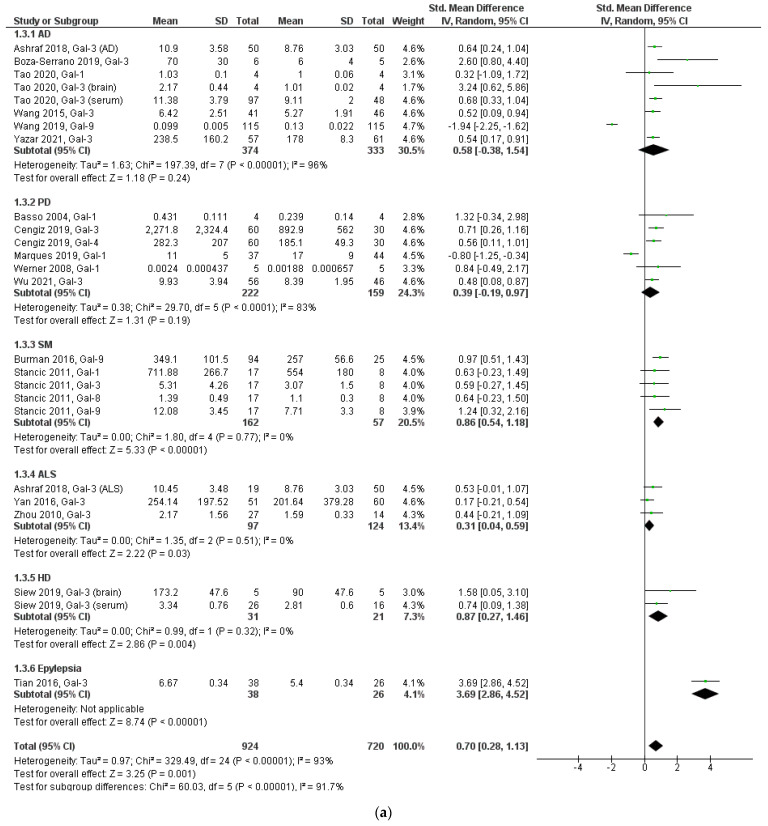
Forest plot of heterogeneity analyses. (**a**) Subgroup analysis by type of neurodegenerative disease, (**b**) subgroup analysis by type of galectin.

**Figure 4 biomolecules-12-01062-f004:**
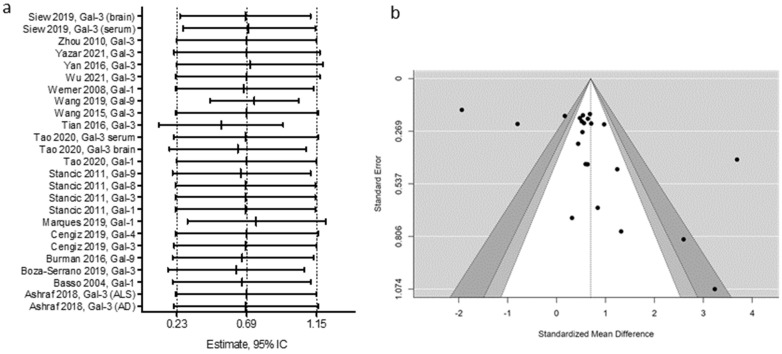
Sensitivity analysis and publication bias. (**a**) Sensitivity plot showing standardized mean difference and 95% confidence interval when excluding one study at a time from meta-analysis. (**b**) Funnel plot for analysis of galectin levels in neurodegenerative diseases.

**Table 1 biomolecules-12-01062-t001:** Characteristics of the included studies.

First Author and Year	Case Nationality	Region	Disease	Number of Patients	Controls	Female % Patients	Median or Mean Age Patients	Median or Mean Age Controls	Study Design	Detected Sample	Assay Method	Galectins Tested	Units
Boza-Serrano, 2019 [11]	Dutch and Swedish	Europe	AD	6	5	16.6	74.6	74.8	CS	Brain	WB	Gal-3	FC
Yazar, 2021 [24]	Turkish	Western Asia	AD	57	61	61.4	79.1	77.7	CS	Serum	ELISA	Gal-3	pg/mL
Cengiz, 2019 [8]	Turkish	Western Asia	PD	60	30	40	72.5	72	CS	Serum	ELISA	Gal-3 and -4	pg/mL
Marques, 2019 [25]	Dutch	Europe	PD	37	44	35	57	58	CS	CSF	ELISA	Gal-1	ng/mg
Wang, 2019 [26]	Chinese	East Asia	AD	115	115	64.3	75.7	75.9	CS	Serum	ELISA	Gal-9	pg/mL
Yan, 2016 [27]	Chinese	East Asia	ALS	51	60	96	54.8	55.5	CS	Plasma	ELISA	Gal-3	ng/mL
Zhou, 2010 [28]	American	America	ALS	27	14	-	-	-	CS	CSF	ELISA	Gal-3	ng/mL
Wang, 2015 [29]	Chinese	East Asia	AD	41	46	46.3	71.2	69.8	CS	Serum	ELISA	Gal-3	ng/mL
Ashraf, 2018 [30]	Saudi	Western Asia	AD	31	50	41.9	66.8	74.9	CS	Serum	ELISA	Gal-3	ng/mL
			ALS	19	-	42.1	64.1		CS	Serum	ELISA	Gal-3	ng/mL
Tian, 2016 [31]	Chinese	East Asia	Epilepsy	38	26	55	32.5	-	CS	Serum	ELISA	Gal-3	ng/mL
Burman, 2016 [32]	Swiss	Europe	MS	94	25	-	46	48		CSF	ELISA	Gal-9	pg/mL
Stancic, 2011 [9]	Dutch	Europe	MS	17	8	-	61	76.6	CS	Brain	WB	Gal-1, -3, -8, and -9	FC
Basso, 2004 [33]	Dutch	Europe	PD	4	4	25	75	70	CS	Brain	2-DE and MALDI-TOF-MS	Gal-1	relative volume in 2-DE
Werner, 2008 [34]	German	Europe	PD	5	5	40	84.2	77.4	CS	Brain	2-DE and MALDI-TOF-MS	Gal-1	relative volume in 2-DE
Tao, 2020 [35]	-	-	AD	101	52	-	-	-	CS	Brain and serum	WB and ELISA	Gal-3 and -1	FC and ng/mL
Wu, 2021 [36]	Taiwanese	East Asia	PD	56	46	43	64.8	65.9	CS	Plasma	ELISA	Gal-3	ng/mL
Siew, 2019 [14]	Taiwanese	East Asia	HD	31	21	72	51	54	CS	Plasma and brain	ELISA and RTqPCR	Gal-3	ng/mL
**Articles included in the qualitative analysis**													
Wada, 2003 [37]			ALS	12	10	33	58.3	60.1	CS	Skin	IHC	Gal-1	pixels
Zubiri, 2018 [10]	British	Europe	ALS	12	6	33			PP	Plasma	LC-MS/MS	Gal-1 and -3	FC
Kato, 2001 [21]	-	-	ALS								IHC	Gal-1	
de Jong, 2018 [22]	-	-	SM	25	11	-	-	-	CS	Brain	IHC	Gal-4	x
Guo, 2017 [23]	Chinese	East Asia	human prion diseases	5	3	-	-	-	CS	Brain	WB	Gal-1	FC

IHC: immunohistochemistry; qRT-PCR: quantitative real-time polymerase chain reaction; WB: Western blot; LC-MS/MS: liquid chromatography tandem mass spectrometry; 2-DE: two-dimensional gel electrophoresis; CS: cross-sectional; PP: prospective; CSF: Cerebrospinal fluid; AD: Alzheimer’s disease; PD: Parkinson’s disease; SM: multiple sclerosis; HD: HD: Huntington disease; gal: galectin; -: not available.

**Table 2 biomolecules-12-01062-t002:** Subgroup analysis of galectin expression levels in neurodegenerative diseases.

Subgroup	Studies	Test for Association			Test for Heterogeneity		Analytical Model
		SMD	95% CI	*p*	I^2^	*p*	
Neurodegenerative disease and galectin type							
AD and Gal-3	6	0.64	[0.45, 0.83]	<0.00001	44%	*p* = 0.11	FEM
PD and Gal-1	3	0.30	[−1.16, 1.77]	0.68	80%	0.006	REM
PD and Gal-3	2	0.58	[0.28, 0.88]	0.0001	0%	0.45	FEM
SM and Gal-9	2	1.03	[0.62, 1.44]	<0.00001	0%	0.61	FEM
Region							
Western Asia	3	0.60	[0.40, 0.79]	<0.00001	0%	0.98	FEM
East Asia	6	0.70	[−0.46, 1.86]	0.24	98%	<0.00001	REM
Europe	6	0.76	[0.10, 1.41]	0.02	82%	<0.00001	REM
American	1	0.44	[−0.21, 1.09]	0.19	NA	NA	NA
Detection method							
ELISA	13	0.50	[−0.03, 1.03]	0.06	95%	<0.00001	REM
Western blot	3	0.87	[0.47, 1.27]	<0.0001	32%	0.18	FEM
2-DE and MALDI-TOF-MS	2	1.03	[−0.01, 2.07]	0.05	0%	0.66	FEM
sample analyzed							
Serum and plasma	10	0.58	[−0.04, 1.19]	0.07	96%	<0.00001	REM
Brain	6	0.93	[0.57, 1.30]	<0.00001	9%	0.36	FEM
CSF	3	0.20	[−0.94, 1.35]	0.73	93%	<0.00001	REM

REM: random-effects model; FEM: fixed-effects model; SMD: standardized mean difference; CI: confidence interva1; CSF: cerebrospinal fluid; AD: Alzheimer’s disease; PD: Parkinson’s disease; SM: multiple sclerosis; 2-DE: two-dimensional gel electrophoresis; NA: not applicable.

## Data Availability

Not applicable.

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
