# Peer review of "Association between Galectin Levels and Neurodegenerative Diseases: Systematic Review and Meta-Analysis"

_biomolecules, 2022, doi:10.3390/biom12081062_

Round 1

Reviewer 1 Report

Prompted by literature data reporting that the content of galectins differs in the blood and brain of patients with neurodegenerative disorders, the Authors conduct a meta-analysis to determine whether patients with neurodegenerative diseases have higher galectin levels than healthy people. The Authors conclude that patients with neurodegenerative illnesses, such as Alzheimer's disease (AD), amyotrophic lateral sclerosis (ALS), Parkinson's disease (PD) and multiple sclerosis (MS) have higher galectin levels than healthy people. Specifically, it is reported that patients with AD, PD, and ALS had higher levels of galectin 3 expression, while patients with MS have higher levels of galectin 9 expressions.

The manuscript is well written and of interest. However, the review should be improved addressing in more detail the biological/pathological role of galectin 3 and galectin 9, which are increased in neurodegenerative disorders. In particular, I couldn't find a description of galectin 9 in the review. What role could it have in neurodegenerative diseases?

Author Response

The manuscript is well written and of interest. However, the review should be improved addressing in more detail the biological/pathological role of galectin 3 and galectin 9, which are increased in neurodegenerative disorders. In particular, I couldn't find a description of galectin 9 in the review. What role could it have in neurodegenerative diseases?

                A: A description of the biological role of galectin 3 and galectin 9 in neurodegenerative diseases has been added in the discussion (page 14 and 15). In addition, information on the general functions of galectins has been added in the introduction (page 2).

Reviewer 2 Report

In the present work, the authors report a systematic review and meta-analysis of the association between galectin levels and the development of neurodegenerative diseases. In particular, out of 22 total selected studies, 5 were included in the qualitative review, while 17 were addressed in the meta-analysis. The systematic review and meta-analysis was carried out using appropriate methodology and relevant criteria. The results are clearly discussed and presented in a manner that is consistent with the evidence. Except for a few typos, the manuscript is well written and sheds light on a biomarker of potentially high importance in the diagnosis and prognosis of neurodegenerative diseases.

A few observations:

The Introduction could be improved. The main galectins involved in neurodegenerative diseases should be described in further detail in this section, and their importance should be further highlighted. . 

Page 2, l. 68: typo - Conchrane instead of Cochrane

Page 2, l.97 - "the data extracted from the selected articles were: two review authors..." it seems like a part of the text was inadvertently deleted, and the resulting paragraph is not quite clear. Please clarify.

Page 4, fig.1: typos - "Records identified throug", "Full-tex articles". Also the number of Full text articles assessed for eligibility (42) does not correspond to the one reported in text (43) and does not make sense in light of the following exclusion step.

Ppage 4, l. 137 typo - it seems like a full-stop and a T went missing ("inflammatory diseases he study")

Page 11, l. 209 - "this because this was where..." this phrasing is convoluted and detracts from the clarity of the sentence, please rephrase.

Page 11, l. 211 typo - "Wester blot.

Apart from these minor points, I recommend this article for publication on Biomolecules.

Author Response

The Introduction could be improved. The main galectins involved in neurodegenerative diseases should be described in further detail in this section, and their importance should be further highlighted.

                A: In the introduction (page 2), we have added information about the functions of galectins in neurodegeneration-related processes and mentioned the involvement of some galectins in neurodegenerative diseases. In addition, information on the specific functions of each galectin in neurodegenerative diseases has been added in the discussion (page 14 and 15).

Page 2, l. 68: typo - Conchrane instead of Cochrane

A: Typo has been corrected

Page 2, l.97 - "the data extracted from the selected articles were: two review authors..." it seems like a part of the text was inadvertently deleted, and the resulting paragraph is not quite clear. Please clarify.

                A: wording has been corrected

Page 4, fig.1: typos - "Records identified throug", "Full-tex articles". Also the number of Full text articles assessed for eligibility (42) does not correspond to the one reported in text (43) and does not make sense in light of the following exclusion step.

                A: figure has been corrected

Ppage 4, l. 137 typo - it seems like a full-stop and a T went missing ("inflammatory diseases he study")

                A: Typo has been corrected

Page 11, l. 209 - "this because this was where..." this phrasing is convoluted and detracts from the clarity of the sentence, please rephrase.

A: wording has been corrected

Page 11, l. 211 typo - "Wester blot.

                A: Typo has been corrected

Round 2

Reviewer 1 Report

The manuscript has been improved and is, in my opinion, suitable for publication.